# An Immunosensor for the Determination of Cathepsin S in Blood Plasma by Array SPRi—A Comparison of Analytical Properties of Silver–Gold and Pure Gold Chips

**DOI:** 10.3390/bios11090298

**Published:** 2021-08-27

**Authors:** Pawel Falkowski, Piotr Mrozek, Zenon Lukaszewski, Lukasz Oldak, Ewa Gorodkiewicz

**Affiliations:** 1Bioanalysis Laboratory, Faculty of Chemistry, University of Bialystok, Ciolkowskiego 1K, 15-245 Bialystok, Poland; pawelfalkowski@wp.pl (P.F.); l.oldak@uwb.edu.pl (L.O.); 2Faculty of Mechanical Engineering, Bialystok University of Technology, Wiejska 45C, 15-351 Bialystok, Poland; p.mrozek@pb.edu.pl; 3Faculty of Chemical Technology, Poznan University of Technology, pl. Sklodowskiej-Curie 5, 60-965 Poznan, Poland

**Keywords:** silver–gold chip, immunosensor, cathepsin S, array surface plasmon resonance imaging, blood plasma, ovarian cancer, liquid biopsy

## Abstract

The array SPR imaging (SPRi) technique is well suited to the determination of biomarkers in body fluids, called liquid biopsy. No signal enhancement or analyte preconcentration is required. With the aim of achieving signal enhancement and lowering the cost of a single determination, the replacement of gold-covered chips by silver–gold chips was investigated. The aim of this work was to investigate the analytical characteristics of a biosensor formed on a Ag/Au chip and to compare them with those of a biosensor formed on a gold chip. A biosensor for the determination of cathepsin S (Cath S) was chosen as an example. The biosensor consisted of the linker cysteamine and an immobilized rat monoclonal antibody specific for cathepsin S. Both biosensors exhibited a Langmuirian response to Cath S concentration, with linear response ranging from LOQ to 1.5 ng mL^−1^. The LOQ is 0.1 ng mL^−1^ for the biosensor formed on the Ag/Au chip, and 0.22 ng mL^−1^ for that formed on the gold chip. Recoveries and precision for medium and high Cath S concentrations were acceptable for both biosensors, i.e., precision better than 10% and recoveries within the range 102–105%. However, the results for the lowest Cath S concentration were better for the biosensor formed on the Ag/Au chip (9.4 and 106% for precision and recovery, respectively). Generally, no significant differences in analytical characteristics were observed between the Ag/Au and Au chips. The two biosensors were also compared in the determination of Cath S in real samples. Nine plasma samples from healthy donors and nine from patients with ovarian cancer were analyzed for Cath S concentration with the biosensors formed on Ag/Au and Au chips. The results obtained with the two biosensors were very similar and show no significant differences on the Bland–Altman plot. The Cath S concentration in the blood plasma of ovarian cancer patients was elevated by one order of magnitude as compared with the control (12.6 ± 3.6 vs. 1.6 ± 1.2 ng mL^−1^).

## 1. Introduction

The development of new biosensors is one of the leading areas of research in modern analytical chemistry. The purpose of biosensors is to make specific determinations of significant biomarkers. The determination of biomarkers in body fluids, such as blood plasma or serum, urine, saliva, and cerebrospinal fluid, is called ‘liquid biopsy’. Liquid biopsy has enormous diagnostic potential in the detection of cancer and cardiovascular or neurodegenerative diseases, which, so far, is only at an early stage of development. A variety of biomarkers, such as CA 125, HE 4, CEA, PSA, Troponins T, I and C, and numerous antibodies against bacterial or viral infections, are used broadly in diagnostics. However, these biomarkers have insufficient diagnostic efficiency, and none of them has a diagnostic accuracy of 100%. On the other hand, medical professionals rely on them to determine the stage of a disease and the effectiveness of treatment. Therefore, new biomarkers and new analytical methods are required. Cathepsin S is an emerging biomarker. Surface Plasmon Resonance (SPR) techniques with suitable biomarkers belong to analytical tools used in liquid biopsy. Among SPR techniques, the array SPR imaging (SPRi) is well suited to this purpose. Biosensors for the determination of cathepsins B, D, G [1], L, and S [2], laminin-5 [3], fibronectin [4], collagen IV [5], proteasome 20S and immunoproteasome 20S, UCHL-1 [6], MMP-1 [7] and MMP-2 [8], aromatase, podoplanin, leptin, CA 125, CEA, HE 4, and cystatin C in body fluids have been developed [9]. Many of these biosensors have been used in clinical investigations, including those of bladder cancer [10,11]. All these biosensors are able to determine the presence of a suitable marker without signal enhancement or preliminary preconcentration. The measurement process in the array SPR imaging technique differs from that commonly used in the fluidic version of SPR. Unlike in the fluidic version of SPR, where a biosensor is manufactured in situ, in the array version of SPRi, a biosensor is manufactured ex situ. The second significant difference is the SPR measurement process: in the fluidic version of SPR, the measurement is performed with liquid present, while in the array version of SPRi, the SPRi measurement is performed after the gentle removal of liquid.

In the array version of SPRi, an array consisting of 108 measuring points is usually used [3,4,5,6,7,8]. These points form nine measurement cells separated by hydrophobic paint, each of them having 12 measuring points (Figure 1). Nine samples are measured simultaneously. The average from the 12 measuring points, after a suitable statistical procedure, is considered as a single result, which ensures satisfactory precision. Prior to the creation of the biosensor, there is bare gold on the bottom of a measuring point.

SPR biosensors are usually based on chips consisting of a thin layer of gold on a glass plate. With the aim of lowering the cost of a single determination, the replacement of gold-covered chips with silver–gold chips may be considered. Apart from gold, the SPR effect is also observed on other metals, including silver [12,13,14,15,16]. The SPR dip with silver is even sharper than with gold. However, a silver surface is susceptible to oxidation, which disturbs biosensor creation. Therefore, silver covered with gold seems to maintain the silver SPR dip and the surface properties of gold [13,14]. Recently, a paper considering the sensitivity of a Ag/Au biosensor has been published [17]. This paper was the inspiration for this paper. The changes in the SPR effect with a chip consisting of 43 nm silver covered with 4 nm of gold in comparison to a chip covered with 50 nm of gold are shown in Figure 2.

The SPR dip of the Ag/Au chip is much sharper than that of the Au chip and is located at a lower angle. Therefore, a biosensor constructed on the Ag/Au chip should react more strongly than a biosensor formed on a Au chip. An advantage of the Ag/Au chip may be its lower manufacturing cost.

The aim of this work was to investigate the analytical characteristics of a biosensor formed on a Ag/Au chip and to compare them with those for a biosensor formed on a gold chip. A recently developed biosensor for the determination of cathepsin S (Cath S) was chosen as an example [2]. This biosensor was used in a previous paper [17] for an initial examination of the analytical potential of this biosensor. The biosensor consists of the linker cysteamine and an immobilized rat monoclonal antibody specific for cathepsin S. Cath S is one of 11 cysteine cathepsin proteases. It consists of 217 amino acids and has an MW of 23.7 kDa. Cath S shows optimal activity at pH values of 6.50–7.50 [18,19,20]. This enzyme is an emerging marker for various cancers [18,19,20,21,22,23], as well as a biomarker for inflammation, insulin resistance, the risk of developing diabetes, and mortality risk [24]. Preliminary experiments show that Cath S can also be an ovarian cancer marker.

## 2. Apparatus, Chip Manufacture, Materials and Methods

### 2.1. SPRi Apparatus

SPRI measurements were performed using a prototype apparatus dedicated to array SPRi measurements developed at the Laboratory of Bioanalysis at the University of Bialystok. The apparatus consists of a diode laser emitting a light beam with a wavelength of 635 nm. The beam is then aligned and narrowed by transmission through a fiber collimator. At the next stage, the beam is broadened through an expander and reflects on a linear polarizer, which is then used to polarize the light beam to two polarization types: p and s. Basic measurements are performed with p polarization, while s polarization is used for measurements of the background. At the top part of the measuring system is a glass prism, with a glass chip placed on it in a Kretschmann configuration. The light is reflected from the chip metallic layer and detected by a CCD digital camera. The signal is then transferred to a computer with graphical software, which allows evaluation of the SPRi images in 2D form. The instrument contains a controlling unit, which enables automatic measurement set-up. A sketch of the apparatus is shown in Figure 3.

### 2.2. Chip Manufacture

#### Deposition of Metallic Cr-Ag-Au Layers onto Glass Substrate

Glass substrates with dimensions 20 × 20 × 1 mm and refractive index *n* = 1.51 were cut out from microscope slides (Thermo Scientific, Waltham, MA, USA). The glass plates were polished using an aqueous suspension of cerium oxide. The surfaces of the substrates were cleaned with the use of detergent, acetone, and isopropyl alcohol. The slides were rinsed and washed ultrasonically in deionized water between every use of a cleaning agent. Thin metallic films were deposited onto the surface of the glass by means of physical vapor deposition in an NA501 vacuum system in a vacuum of 8 × 10^−6^–1 × 10^−5^ hPa at an ambient temperature. Molybdenum boats were used as resistive evaporation sources. The glass plates were placed on a rotary substrate holder. In the first step, 1 nm of an adhesive Cr layer (99.9%) was deposited at a rate of approx. 0.1 nm/s. Next, approx. 42 nm of Ag (99.99%) was deposited at a rate of 0.08 nm/s, and approx. 5 nm of Au (99.99%) at a rate of 0.01 nm/s. The layer thickness and deposition rate were monitored by means of a quartz crystal microbalance. A sketch of the structure of the Ag/Au chip substrate is shown in Figure 4.

The thickness of 50 nm of Au on a glass substrate is a well-established standard in SPR measurements. The thickness of Ag and Au was determined in preliminary experiments.

### 2.3. Preparation of Separating Paint Layers

In the first step, the clean, unprinted Ag/Au chip substrate underwent a thorough and precise cleaning process by repeated rinsing with dichloroethane and redistilled water and drying with argon between rinses. A light-sensitive paint coating, Elpemer SD 2047 (Lackwerke Peters GmbH & Co. KG, Kempen, Germany), the main component of which was novolac epoxy acrylate, was applied on the cleaned surface, to form a blocking polymer layer. A flat screen with a mesh density of 90 lpc and a paint with appropriate viscosity were used for this purpose. The printed glass plate was then dried in a convection oven at 65 °C for 60 min. To obtain appropriate measuring cells (a group of measuring points, not containing photosensitive paint), a transparent foil mask with black opaque points 0.40 mm in diameter was placed on the printed and dried plate. The prepared plate was sensitized with UV light, with an output power of 48 W, for 90 s. In the next step, the irradiated photopolymer layer was treated by spraying with a 1% sodium carbonate solution and rinsing with redistilled water, then dried under a stream of argon. In places where the photopolymer was covered by a black mask, not exposed to a UV light, a clean layer of metal was exposed after rinsing. These raw metal fields were used as measuring points. The next step in the preparation of the chip was the printing of a blue hydrophobic polymer layer, which was the border separating individual groups of measurement points and preventing the mixing of analyte solutions. The hydrophobic layer was also applied by a screen-printing technique, using a screen with a mesh of 120 L/cm. The blue ink layer, Elpemer SD 2457 (Lackwerke Peters GmbH & Co. KG, Germany), was printed on a pre-printed plate, and then the chip was dried again in a convection oven at 65 °C for 60 min, after which it was irradiated with UV light for 5 min. A chip printed and dried in this way becomes chemically resistant to the organic solvent, ethanol, used in the subsequent stages of biosensor preparation. Separation of paint layers on the Au chip substrate was performed in the same manner.

### 2.4. Reagents

Cathepsin S protein and a rat monoclonal antibody specific for cathepsin S (R&D Systems, Minneapolis, MN, USA), cysteamine hydrochloride, *N*-ethyl-*N*’-(3-dimethylaminopropyl) carbodiimide (EDC), human albumin (all Sigma, Steinheim, Germany) and N-hydroxysuccinimide (NHS) (Aldrich, Munich, Germany) were used, and absolute ethanol (POCh, Gliwice, Poland), HBS-ES solution pH = 7.4 (0.01 M HEPES, 0.15 M sodium chloride, 0.005% Tween 20, 3 mM EDTA), acetic buffer pH = 3.79–5.57, Phosphate Buffered Saline (PBS) pH = 7.4, phosphate buffer pH = 7.17–8.04, acetic buffer pH = 3.79–5.57, Phosphate Buffered Saline (PBS) pH = 7.4, phosphate buffer pH = 7.17–8.04, photopolymer ELPEMER SD 2054, blue ink Elpemer SD 2457 (Lackwerke Peters GmbH & Co. KG, Kempen, Germany) were used as received. Aqueous solutions were prepared with miliQ water (Simplicity^®^ MILLIPORE, Merck KGaA, Darmstadt, Germany).

### 2.5. Biological Material

The Regional Blood Donation and Blood Treatment Center in Bialystok provided the plasma samples of healthy donors, and the Maria Sklodowska-Curie Oncology Center provided samples from patients with ovarian cancer. All samples were provided after obtaining the consent of the Bioethical Commission.

### 2.6. Procedure of Antibody Immobilization

First, the Ag/Au and Au chips were covered with a cysteamine linker. Antibody immobilization was performed following the well-known EDS/NHS protocol, described in a previous paper [2], with the rat monoclonal antibody specific for cathepsin S. 

### 2.7. SPRi Measurement

The selection of the correct SPR angle was the first stage of measurement performed for the chip after the immobilization of the receptor layer. The SPR signal was measured at a fixed SPR angle on the basis of recorded images. The first photograph was taken after the immobilization of the antibody on the biosensor surface. Then, the analyzed solution was applied to the biosensor surface for 10 min. After interaction, the surface of the biosensor was washed with HBS-ES buffer and distilled water and dried under a stream of argon. A second photograph was taken after the drying of the biosensor’s surface. The contrast values obtained for all the pixels across a particular sample spot were integrated. Then, the SPRi signal was integrated over the spot area. The evaluation of the SPR images in two-dimensional form and conversion of numerical signals to a quantitative signal were performed using the National Institutes of Health (NIH) ImageJ software, version 1.32. A calibration graph was used for the evaluation of the cathepsin S concentration.

## 3. Results and Discussion

### Analytical Characteristics of the Biosensor for the Determination of Cathepsin S Formed to the Ag/Au Chip Substrate

To compare the analytical characteristics of the biosensor for the determination of cathepsin S formed on the Ag/Au chip and that formed on the Au chip, the following characteristics were obtained: dependence of the analytical signal on cathepsin S concentration, linear response range, precision and recoveries of cathepsin S determination under model conditions, and limits of detection and quantification. The dependence of the analytical signal on the analyte concentration was examined within the range 0.1–3.5 ng mL^−1^. The optimum concentration of the rat monoclonal antibody specific for cathepsin S was selected in a previous study [2], and was fixed at 20 ng mL^−1^ with pH = 7.4. The results are shown in Figure 5.

Both calibration graphs exhibit Langmuirian-type dependence, which is caused by gradual saturation of active antibodies, and both have linear sections within the range 0.1–1.5 ng mL^−1^. The calibration graph obtained with the biosensor formed on the Ag/Au chip has a slightly greater slope and higher intercept than that for the Au chip. Linearity of the analytical signal shown in the previous paper [17] was confirmed, and the slope and intercept were verified.

The precision and recoveries of the determination of cathepsin S using the biosensors formed on the Ag/Au chip and the Au chip were investigated under model conditions using three spikes of cathepsin S: 0.1, 0.5, and 1.0 ng mL^−1^. The spikes represent the lowest, medium and high concentrations of Cath S. The results for the biosensors formed on the Ag/Au chip and the Au chip are shown in Table 1A,B, respectively. Generally, the recoveries and precision for medium and high Cath S concentrations are acceptable for both biosensors: the precision is better than 10%, and recoveries are within the range 102–105%. However, the results for the lowest Cath S concentration are better for the biosensor formed on the Ag/Au chip (9.4 and 106% for precision and recovery, respectively, compared with 24 and 90% for the biosensor formed on the gold chip). For the biosensor formed on the Ag/Au chip, the limit of detection calculated on the basis of 3SD is equal to 0.03 ng mL^−1^, and the limit of quantification (10SD) is 0.1 ng mL^−1^, while the respective values for the biosensor formed on the Au chip are 0.066 and 0.22 ng mL^−1^. Thus, the accessible linear response range is 0.1–1.5 ng mL^−1^ in the case of the biosensor formed on the Ag/Au chip, and 0.22–1.5 ng mL^−1^ in the case of the biosensor formed on the Au chip. Generally, the differences in the analytical characteristics of the two biosensors are negligible, and basically confirm preliminary data shown in the previous paper [17].

The most significant comparison of the biosensors formed on the Ag/Au and Au chips concerns the determination of Cath S in real samples. Nine plasma samples from healthy donors and nine samples from patients with ovarian cancer were analyzed for Cath S concentration using the two biosensors. Samples with concentrations above the linearity range were diluted with PBS buffer. The results are shown in Table 2 and Table 3, for healthy donors and ovarian cancer patients, respectively.

The average value of Cath S concentration for healthy volunteers was determined to be 1.55 ± 0.91 using the biosensor formed on the Ag/Au chip, and 1.41 ± 0.93 using the biosensor formed on the Au chip.

The average value of Cath S concentration for patients suffering with ovarian cancer was determined to be 12.6 ± 3.2 using the biosensor formed on the Ag/Au chip, and 12.7 ± 4.0 using the biosensor formed on the Au chip.

Although the results obtained using the biosensors formed on different chips seem very similar, their equivalence was checked by means of a Bland–Altman plot, as shown in Figure 6.

The mean value of the difference between the results determined by the biosensors formed on Ag/Au and Au chips is almost equal to zero, and only two results are slightly below the confidence limit. Thus, the results obtained by the biosensors based on different chips are equivalent. It is noteworthy that the Ag/Au chips are significantly cheaper. The results confirm the significantly elevated concentration of cathepsin S in the blood plasma of patients with ovarian cancer.

## 4. Conclusions

A biosensor for the determination of cathepsin S that consists of the linker cysteamine and the immobilized rat monoclonal antibody specific for cathepsin S, formed on a Ag/Au chip, exhibits a Langmuirian response to the Cath S concentration, with a linear response that ranges from 0.1 (LOQ) to 1.5 ng mL^−1^. An analogous biosensor formed on gold chip exhibits a linear response that ranges from 0.22 (LOQ) to 1.5 ng mL^−1^. Both biosensors achieve precision greater than 10% and recoveries within the range 102–105%. Generally, no significant differences in analytical characteristics were observed between the Ag/Au and Au chips. The results for the determination of cathepsin S in plasma samples from patients with ovarian cancer and from healthy donors, as determined by biosensors formed on Ag/Au and Au chips, are very similar and show no significant differences on the Bland–Altman plot. The Cath S concentration in the blood plasma of ovarian cancer patients is elevated by one order of magnitude as compared with the control (12.6 ± 3.6 vs. 1.6 ± 1.2 ng mL^−1^).

## Figures and Tables

**Figure 1 biosensors-11-00298-f001:**
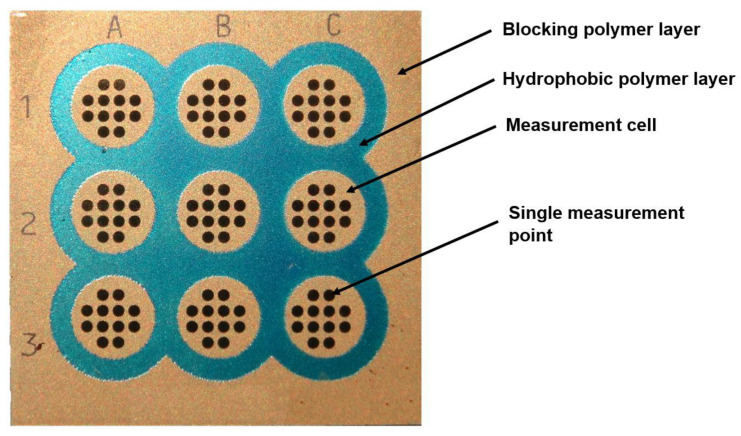
Architecture of chip in the array SPRi technique.

**Figure 2 biosensors-11-00298-f002:**
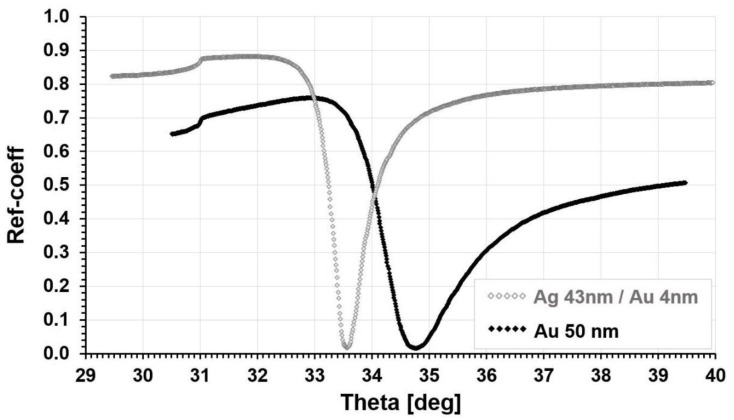
Comparison of the SPR effect with a chip covered with 43 nm of silver and 4 nm of gold and with a chip covered with 50 nm of gold (adapted from [17]).

**Figure 3 biosensors-11-00298-f003:**
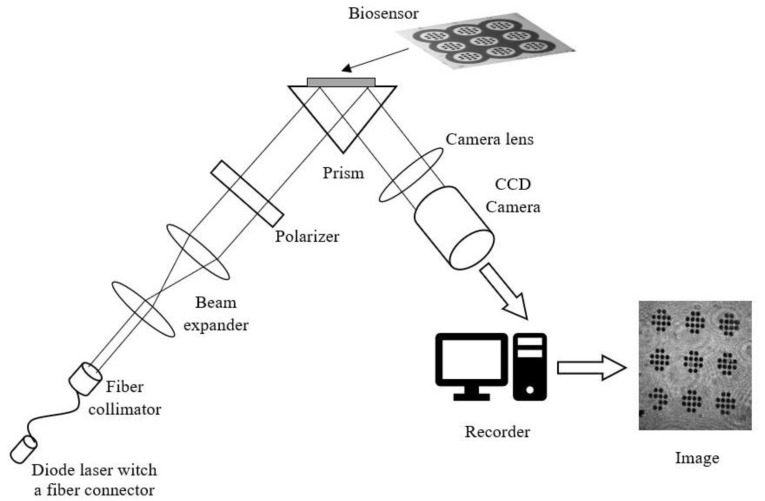
Sketch of SPRi apparatus.

**Figure 4 biosensors-11-00298-f004:**
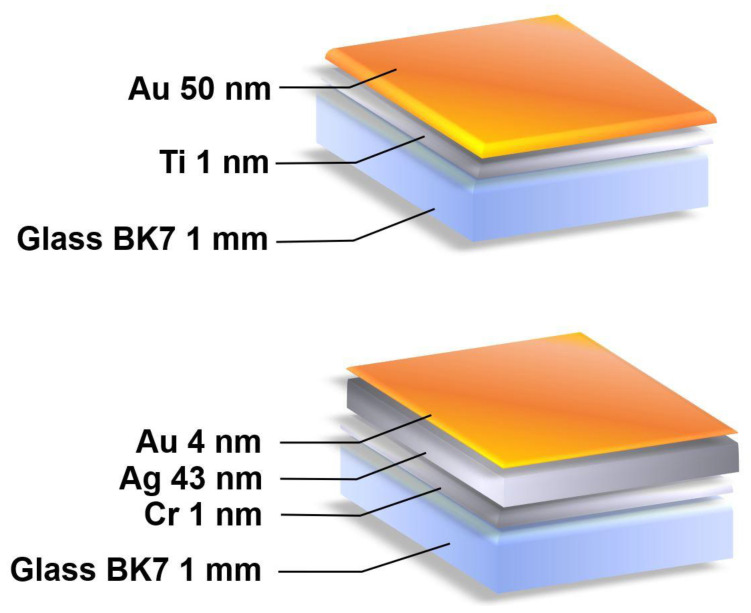
Structure of Ag/Au chip substrate (lower picture) as compared with Au substrate (upper picture).

**Figure 5 biosensors-11-00298-f005:**
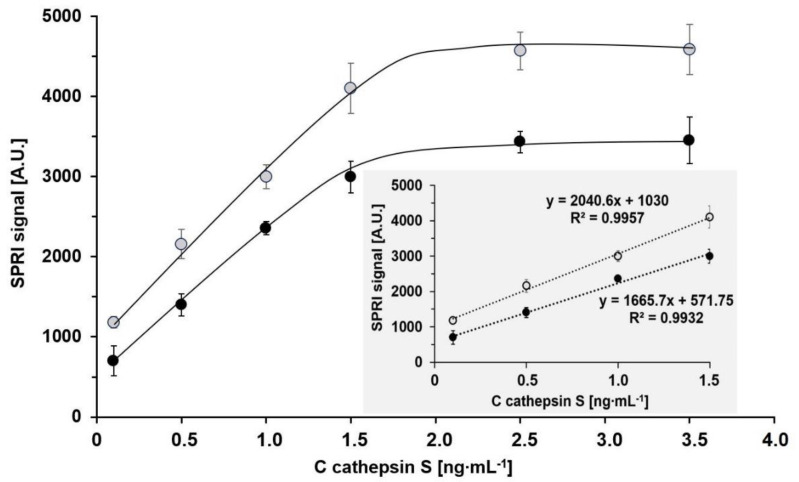
Calibration graphs for cathepsin S with biosensors formed on Ag/Au (open circles) and Au (closed circles) chips. Antibody concentration 20 ng mL^−1^ and pH = 7.4.

**Figure 6 biosensors-11-00298-f006:**
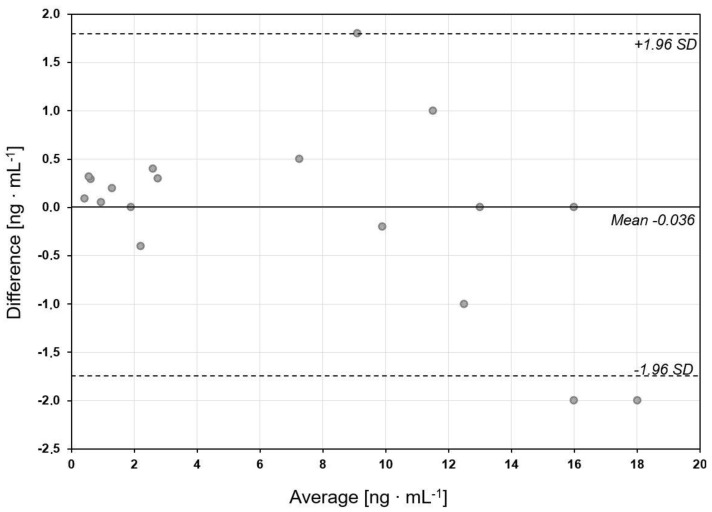
Bland-Altman plot for comparison of the results of cathepsin S in blood plasma as determined by biosensors formed on Ag/Au and Au chips.

**Table 1 biosensors-11-00298-t001:** **A.** Precision and recoveries of the determination of cathepsin S with the biosensor formed on the Ag/Au chip (*n* = 24). **B.** Precision and recoveries of the determination of cathepsin S with the biosensor formed on the Ag/Au chip (*n* = 24).

**(A)**			
**Spike [ng mL^−1^]**	**Found [ng mL^−1^]**	**Recovery [%]**	**RSD [%]**
0.100	0.106	105.9	9.4
0.500	0.519	103.8	3.8
1.000	1.015	101.5	7.3
**(B)**			
**Spike [ng mL^−1^]**	**Found [ng mL^−1^]**	**Recovery [%]**	**RSD [%]**
0.100	0.090	90.2	24.0
0.500	0.512	102.3	2.0
1.000	1.049	104.9	1.9

**Table 2 biosensors-11-00298-t002:** Cathepsin S in blood plasma samples of healthy volunteers as determined by biosensors formed on the Ag/Au and Au chips.

Sample No.	Chip Ag/Au		Chip Au	
	Cath S [ng mL^−1^]	SD [ng mL^−1^]	Cath S [ng mL^−1^]	SD [ng mL^−1^]
1	0.48	0.05	0.39	0.03
2	0.98	0.04	0.93	0.01
3	2.90	0.07	2.60	0.03
4	1.90	0.04	1.90	0.03
5	0.77	0.04	0.48	0.03
6	1.40	0.04	1.20	0.03
7	2.80	0.11	2.40	0.04
8	2.00	0.03	2.40	0.02
9	0.72	0.09	0.40	0.02
average	1.9	1.5	1.4	0.9

**Table 3 biosensors-11-00298-t003:** Cathepsin S in blood plasma samples of ovarian cancer patients as determined by biosensors formed on Ag/Au and Au chips.

Sample No.	Chip Ag/Au		Chip Au	
	Cath S [ng mL^−1^]	SD [ng mL^−1^]	Cath S [ng mL^−1^]	SD [ng mL^−1^]
18A	16.0	0.32	16.0	0.68
19A	17.0	0.24	19.0	0.80
21A	12.0	0.65	13.0	0.49
23A	12.0	0.24	11.0	0.27
24A	15.0	0.22	17.0	0.67
25A	9.8	0.41	10.0	0.17
26A	10.0	0.12	8.2	0.13
27A	13.0	0.30	13.0	0.54
28A	7.5	0.13	7.0	0.11
average	12.5	3.1	12.7	4.1

## Data Availability

Not applicable.

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
