# Peer review of "An Immunosensor for the Determination of Cathepsin S in Blood Plasma by Array SPRi—A Comparison of Analytical Properties of Silver–Gold and Pure Gold Chips"

_biosensors, 2021, doi:10.3390/bios11090298_

Round 1
Reviewer 1 Report
The manuscript reports a study comparing a SPRi sensor performance for detection of cathepsin S, an ovarian cancer biomarker, in blood plasma with the use of regular gold coated sensor chip and silver/gold hybrid coating. The conclusion is that no significant performance difference between the two types of sensor chips for detection of cathepsin S, although the silver/gold coated chip has slightly better LOQ and low concentration analyte detection capability due to sharper silver resonance dip. The SPRi system and the sensor array has been reported multiple times previously, but the comparison study on silver/gold chip is new for this system. The study is well designed and reported, with sufficient data supporting the conclusion. The subject is fit well to the special issue. However, the significance and motivation of the study, and the use of a sensor array is not well justified, as detailed below. These questions need to be addressed to justify the publication value of the paper.
- The motivation of this work is assuming switching from gold coated sensor chip to silver/gold hybrid sensor chip will reduce the cost of the assay, but this is not well justified. What is the estimated cost saving by switch gold film to silver/gold film in term of percentage of total assay cost? The hybrid film may increase the labor and instrumentation cost on the fabrication. Will the saving on the cost of gold cover it?
- What is the value of using 12 sensor spots in each hydrophobic region if they are functionalized with the same probe and detecting the same analyte? Averaging of the 12 sensor spot data will be equivalent to just functionalize the whole area uniformly with a large spot.
- Source of the photo sensitive polymer materials for preparing the sensor chips was not provided.
Author Response
Answers to Reviewer’s #1 comments
The authors would like to express their thanks for the very helpful and stimulating comments.
1.The motivation of this work is assuming switching from gold coated sensor chip to silver/gold hybrid sensor chip will reduce the cost of the assay, but this is not well justified. What is the estimated cost saving by switch gold film to silver/gold film in term of percentage of total assay cost? The hybrid film may increase the labor and instrumentation cost on the fabrication. Will the saving on the cost of gold cover it?
The cost of materials was a significant motivation for this work, but not the only one. We also expected a stronger signal with Ag/Au, which was found to some extent. As regards the cost, the current price of gold is approximately $1850 per ounce, while the silver price is only approximately $ 25 per ounce. A greater variety of material solutions for use chips is also a significant advantage. In the light of this comment, we have modified the text slightly (abstract line 18, introduction, line 99).
2.What is the value of using 12 sensor spots in each hydrophobic region if they are functionalized with the same probe and detecting the same analyte? Averaging of the 12 sensor spot data will be equivalent to just functionalize the whole area uniformly with a large spot.
Assuming the homogeneity of the whole area, in terms of antibody distribution and cathepsin S capture, the Reviewer’s argument is correct. However, due to inhomogeneity, the array of measuring points, after a suitable statistical procedure, gives significantly better results. An explanation of this has been added (line 74).
3.Source of the photo sensitive polymer materials for preparing the sensor chips was not provided.
The suggested changes have been introduced (lines 158 and 174-175)
Reviewer 2 Report
This paper suggested that the array version of SPR spectrometer is better method than the conventional SPR spectrometer in two points, such as the array type detection and Ag/Au chip. This concept presents the novel spectroscopic tool for detecting specific biomolecules. I suggest the minor correction for improving this paper.
- The x-axis in Figure5 is missing "4.0".
- If you change the Table 2 into the graph for efficient comparison between two different Chips, it would be great for understanding the efficient chip design.
- I could not find the conclusion part in manuscript. If you miss the conclusion part, please add this part in your manuscript.
Author Response
Answers to Reviewer’s #2 comments
We would like to express our thanks to the Reviewer for his very encouraging comments
1.The x-axis in Figure5 is missing "4.0".
The suggestion has been introduced.
2.If you change the Table 2 into the graph for efficient comparison between two different Chips, it would be great for understanding the efficient chip design.
We believe that Tables 2 and 3 give the reader an opportunity to compare each of 18 pairs of results.
3.I could not find the conclusion part in manuscript. If you miss the conclusion part, please add this part in your manuscript
Conclusions have been introduced, as suggested (lines 306-319)
Reviewer 3 Report
The authors consider impact of metallic support to parameters of SPRi-based biosensors. The obtained results are not associated with significant improvements of these biosensors, but could be useful for further studies in this field. Besides, data presentation and discussion need substantial revision. Nevertheless, the manuscript after its improvements can be published in the Biosensors journal.
The following revisions are recommended:
- Abstract: The values «better that 10%» and «9.4%» cannot be a subject for grounded comparison.
- Lines 34-35. To compare the 12.6 and 1.5 ng mL-1 as statistically relevant values, the RSDs for the both groups should be added.
- Line 47-49. «A limited number of biomarkers, such as CA 125, HE 4, CEA and PSA, are used broadly in diagnostics. » The statement is strange as well as diagnostic control of various cardiac markers is a common practice for acute infarction treatment, and dozens of oncomarkers are actively controlled. Moreover, anti-viral and anti-bacterial antibodies are also actively monitoring biomarkers of various diseases (see the current pandemic situation).
- The choice of SPR techniques as preferable tools for biomarkers control (lines 53-54) should be grounded.\
- Lines 63-64: «Unlike the fluidic version of SPR, where a biosensor is created in situ, in the array version of SPRi, a biosensor is created ex situ». The statement is strange. Biosensors are manufactured before their use. How the authors differentiate «manufacturing» and «creation»?
- Lines 68-69. «In the array version of SPRi, an array consisting of 108 measuring points is usually used.» This geometry (9x12) is a choice of some specific manufacturer. This number does not follow from basic principles of SPRi. At whole, the authors should focus their description on some named technical realization of SPRi sensor indicating its manufacturer and conventional format but not to expand their statements to all possible SPRi sensors.
- Line 82: «Biosensors are usually based on chips containing a thin layer of gold on a glass plate». The authors again extend the experience about SPR sensors to all existing variety of biosensoric devices.
- The Introduction should clearly separate description of previous studies and aims of the presented investigation. Actually the status of Figs. 1, 2 from this point of view is not clear. Moreover, the authors have recently published the paper «Two SPRi biosensors for the determination of cathepsin S in blood plasma» (Talanta, 2021, see ref. [2]), where the same instrumental device was used to detect the same analyte. This background should be clarified in detail to understand the following addressing to already proposed/optimized techniques.
- Section 2.1. If «a prototype apparatus dedicated to array SPRi measurements developed at the Laboratory of Bioanalysis at the University of Bialystok» was described in earlier publication(s), the corresponding reference(s) should be given. Similarly, if chip manufacturing (Sections 2.2, 2.3) was earlier described, the corresponding reference(s) should be given.
- Was the authors’ decision to prepare Ag/Au support for SPRi sensors the first one or some predecessors can be found in literature? This is very important issue that should be clearly commented.
- To compare Au and Ag/Au supports for SPRi sensors, the authors have manufactured two kinds of chips with some chosen thickness of their layers – see Fig. 4. However, the reasons of this choice are not clear from the manuscript. What will be the possible consequences of changing these parameters?
- The title of the Section 2.6 («Procedures») seems strange. Apparently, the authors describe the preparation of chips for measurements.
- The Section 2.4 «Reagents» should be moved to the beginning or end of «Materials and Methods», and the indication of all reagents used (except for the components of ordinary working solutions) in it should be checked.
- The data presented at Fig. 5 should be processed to find statistically significant detection limits (basing on three-sigma rules). What values of the signal were recorded in the absence of cathepsin S?
- Which samples were spiked to obtain the data given in Tables 1A, 1B? Plasma samples without cathepsin S or something else? (Pay attention that the comparison of spiked buffer with the calibration curve in the same buffer cannot characterize recovery of assay. On another hand, as follow from Table 2, the obtaining cathepsin S-free plasma needs additional treatment of common preparations from donors.) What means n=24 for the presented data? 24 independently prepared and tested samples or repeated measurements for some smaller quantity of samples?
- Lines 247-248. «Generally, the differences in the analytical characteristics of the two biosensors are negligible.» The presented data really confirm this statement. So this basic conclusion should be clearly formulated in the finalizing remarks of the article and in the Abstract. Actually the Abstract contains a lot of quantitative data for the both compared chips, but really all or almost all of them do not differ from statistical point of view. This fact should be clearly formulated.
- How the knowledge about the presence or absence of ovarian cancer for the tested persons was confirmed? Have the cases of other diseases associated with changes in the cathepsin S level been tested and excluded? ((See the next note.)
- The cathepsin S is known as biomarker of several diseases and dysfunction, first of all associated with inflammation (see https://uu.diva-portal.org/smash/get/diva2:757170/FULLTEXT01.pdf as an example). This fact should be indicated in the revised manuscript.
- How the found values of cathepsin S accord to literature data about healthy persons and patients with ovarian cancer?
- The calibration curves presented in the paper (Fig. 5) demonstrate saturation at approx. 1.5 ng mL-1 of cathepsin S. So to measure such values as 10-17 ng mL-1, the samples should be diluted to some suitable number of times. How this work was made?
- The values in Table 3 demonstrate RSD for the presented measurements in the range 1.2-5.5% that is much lower as compared with the analytical characteristics stated above (near 10%). Please check and clarify these results.
Author Response
Answers to Reviewer’s #3 comments
The authors express their thanks for the very thorough and stimulating review, as well as for the indication of a very valuable source of data concerning Cathepsin S.
1.Abstract: The values «better that 10%» and «9.4%» cannot be a subject for grounded comparison.
In our opinion, there is no controversy in the abstract. The precision of measurement depends on the measured analyte concentration and becomes worse at the boundary of the method ability. This is the case of the Ag/Au chip with the concentration 0.1 ng mL-1.
2.Lines 34-35. To compare the 12.6 and 1.5 ng mL-1 as statistically relevant values, the RSDs for the both groups should be added.
The suggested corrections have been introduced (lines 36-37).
3.Line 47-49. «A limited number of biomarkers, such as CA 125, HE 4, CEA and PSA, are used broadly in diagnostics. » The statement is strange as well as diagnostic control of various cardiac markers is a common practice for acute infarction treatment, and dozens of oncomarkers are actively controlled. Moreover, anti-viral and anti-bacterial antibodies are also actively monitoring biomarkers of various diseases (see the current pandemic situation).
We partially agree with this comment. Therefore, suitable changes have been introduced (lines 49-51). However, the huge potential body fluids, as a source of diagnostic information, is used to a negligible degree.
4.The choice of SPR techniques as preferable tools for biomarkers control (lines 53-54) should be grounded.
We believe, that such an explanation is contained in the text following this sentence. About 50 papers have been published by our group concerning array SPRi in liquid biopsy.
5.Lines 63-64: «Unlike the fluidic version of SPR, where a biosensor is created in situ, in the array version of SPRi, a biosensor is created ex situ». The statement is strange. Biosensors are manufactured before their use. How the authors differentiate «manufacturing» and «creation»?
We agree with this comment, and suitable changes have been introduced (lines 66-67).
6.Lines 68-69. «In the array version of SPRi, an array consisting of 108 measuring points is usually used.» This geometry (9x12) is a choice of some specific manufacturer. This number does not follow from basic principles of SPRi. At whole, the authors should focus their description on some named technical realization of SPRi sensor indicating its manufacturer and conventional format but not to expand their statements to all possible SPRi sensors.
To date, the array version of SPRi is used only by our group. This is not to be extended to the SPRi technique as a whole.
7.Line 82: «Biosensors are usually based on chips containing a thin layer of gold on a glass plate». The authors again extend the experience about SPR sensors to all existing variety of biosensoric devices.
We agree with this comment. Therefore, more precise ‘SPR biosensors…..’ has been introduced (line 85).
8.The Introduction should clearly separate description of previous studies and aims of the presented investigation. Actually the status of Figs. 1, 2 from this point of view is not clear. Moreover, the authors have recently published the paper «Two SPRi biosensors for the determination of cathepsin S in blood plasma» (Talanta, 2021, see ref. [2]), where the same instrumental device was used to detect the same analyte. This background should be clarified in detail to understand the following addressing to already proposed/optimized techniques.
We cannot agree with this proposal. The architecture of the chip will not be clear without Fig. 1, and the reason for attempting to replace an Au chip with an Ag/Au chip will not be clear without Fig. 2. We expected a stronger SPRi signal, as well as the provision of more options, including in terms of the cost of measurement.
9.Section 2.1. If «a prototype apparatus dedicated to array SPRi measurements developed at the Laboratory of Bioanalysis at the University of Bialystok» was described in earlier publication(s), the corresponding reference(s) should be given. Similarly, if chip manufacturing (Sections 2.2, 2.3) was earlier described, the corresponding reference(s) should be given.
A description of the prototype apparatus has not previously been published. A references concerning chip manufacture has been added ([3-8]).
10.Was the authors’ decision to prepare Ag/Au support for SPRi sensors the first one or some predecessors can be found in literature? This is very important issue that should be clearly commented.
We agree with this comment. We have indicated clearly the papers being an inspiration for this paper (line 91).
11.To compare Au and Ag/Au supports for SPRi sensors, the authors have manufactured two kinds of chips with some chosen thickness of their layers – see Fig. 4. However, the reasons of this choice are not clear from the manuscript. What will be the possible consequences of changing these parameters?
The thickness of 50 nm of Au on a glass substrate is a well established standard in SPR measurements. As for the Ag/Au chip, the thickness of Ag and Au was determined in preliminary experiments. A more thorough answer would constitute a separate problem.
12.The title of the Section 2.6 («Procedures») seems strange. Apparently, the authors describe the preparation of chips for measurements.
We agree with this comment. The subtitle has been changed to ‘Section 2.6. Procedure of antibody immobilization ‘
13.The Section 2.4 «Reagents» should be moved to the beginning or end of «Materials and Methods», and the indication of all reagents used (except for the components of ordinary working solutions) in it should be checked.
We have changed the subtitle to ‘Apparatus, Chip Manufacture, Materials and Methods’. The description of the materials has been supplemented (lines 188-195).
14.The data presented at Fig. 5 should be processed to find statistically significant detection limits (basing on three-sigma rules). What values of the signal were recorded in the absence of cathepsin S?
We agree with this comment. However, we believe that such considerations are given in lines 246-262. Standard deviation of blank is equal to 0.01 ng mL-1 for the Ag/Au biosensor and 0.022 ng mL-1 for the Au biosensor. LOD was calculated as 3SD and LOQ as 10 SD.
15.Which samples were spiked to obtain the data given in Tables 1A, 1B? Plasma samples without cathepsin S or something else? (Pay attention that the comparison of spiked buffer with the calibration curve in the same buffer cannot characterize recovery of assay. On another hand, as follow from Table 2, the obtaining cathepsin S-free plasma needs additional treatment of common preparations from donors.) What means n=24 for the presented data? 24 independently prepared and tested samples or repeated measurements for some smaller quantity of samples?
Tables 1 A and 1B show data concerning a model investigation in PBS buffer. ‘n=24’ means 24 measurements for the same model sample located at 2 x 12 measuring points. We agree that an investigation of recovery with a real plasma sample would be better.
16.Lines 247-248. «Generally, the differences in the analytical characteristics of the two biosensors are negligible.» The presented data really confirm this statement. So this basic conclusion should be clearly formulated in the finalizing remarks of the article and in the Abstract. Actually the Abstract contains a lot of quantitative data for the both compared chips, but really all or almost all of them do not differ from statistical point of view. This fact should be clearly formulated.
The recommended conclusions have been added in the Conclusions (lines 313-314 and Abstract (lines 30-31).
17.How the knowledge about the presence or absence of ovarian cancer for the tested persons was confirmed? Have the cases of other diseases associated with changes in the cathepsin S level been tested and excluded? ((See the next note.)
Plasma ovarian cancer samples were provided by an oncologyst with whom we co-operate in a different area. However, the paper is analytical in nature, and the samples served as an example of the potential of the biosensor, not for clinical investigation.
18.The cathepsin S is known as biomarker of several diseases and dysfunction, first of all associated with inflammation (see https://uu.diva-portal.org/smash/get/diva2:757170/FULLTEXT01.pdf as an example). This fact should be indicated in the revised manuscript.
We have gladly introduced this reference ([20]), together with information concerning cathepsin S as a biomarker of inflammation, insulin resistance, the risk of developing diabetes, and mortality risk (lines 107-108).
19.How the found values of cathepsin S accord to literature data about healthy persons and patients with ovarian cancer?
We have no data for such a comparison. In our opinion, such a comparison would be necessary for clinical investigation.
20.The calibration curves presented in the paper (Fig. 5) demonstrate saturation at approx. 1.5 ng mL-1 of cathepsin S. So to measure such values as 10-17 ng mL-1, the samples should be diluted to some suitable number of times. How this work was made?
Samples with concentration above the linearity range were dilluted with PBS buffer. We have frequently used such an apporoach with other developed biosensors.
21.The values in Table 3 demonstrate RSD for the presented measurements in the range 1.2-5.5% that is much lower as compared with the analytical characteristics stated above (near 10%). Please check and clarify these results.
We agree with this observation. Usually the biosensors used in our technique (array SPRi) exhibit better precision with natural samples than those in the model investigation. We have no clear explanation for this fact. It may be caused by the attempt to use an optimum calibration curve when determining natural samples, while the model experiments are performed at extremely low and extremely high concentrations.
Round 2
Reviewer 3 Report
At whole, the manuscript has been successfully revised and improved. Only some questions still need additional discussion.
Question 11.The authors have commented the choice of thickness for Au and Ag/Au supports in their answer to the reviewer. However, for readers of the paper this choice is still non-grounded.
Question 14. The manuscript contains very little data of measurements for low concentration of cathepsin S. Fig. 4 presents a point for 0.5 ng mL-1 of cathepsin S ad only one lower point for approx 0.05-0.1 ng mL-1. Even zero point is not presented. In this situation the authors’ conclusions about LOD and LOQ need in detailed justification. Please give average zero (blank) signal and its SD in the manuscript. Basing on these data, the LOD and LOQ values will be not only reported, but also substantiated. The actually given clarification is strange: «Standard deviation of blank is equal to 0.01 ng mL-1 for the Ag/Au biosensor and 0.022 ng mL-1 for the Au biosensor. LOD was calculated as 3SD and LOQ as 10 SD.» SD is measured in A.U. of SPRi signals, whereas LOD and LOQ are measured in ng mL-1. Parameters with different dimensions cannot be equal.
Question 19 was the following: «How the found values of cathepsin S accord to literature data about healthy persons and patients with ovarian cancer?». The authors have answered that «We have no data for such a comparison». However, even in the initial manuscript they said that «Cath S … This enzyme is an emerging marker for ovarian cancer». Thus, somebody compared earlier the Cath S levels for healthy persons and for patients with ovarian cancer and made the conclusion that cathepsin S is an oncomarker (that was then included in the manuscript). The reviewer is asking only to present data of this somebodies' study of Cath S levels.
Question 20. As well as the necessary dilution is not known, the analyst should test several dilutions to to leave the saturation level (plateau) of the calibration curve. This fact should then be reflected in the description of the analysis method.
Author Response
Answers to Reviewer’s comments
The authors thanks the reviewer for a very interesting and stimulating discussion
Question 11.The authors have commented the choice of thickness for Au and Ag/Au supports in their answer to the reviewer. However, for readers of the paper this choice is still non-grounded.
The recommended information has been introduced (lines 157-158).
Question 14. The manuscript contains very little data of measurements for low concentration of cathepsin S. Fig. 4 presents a point for 0.5 ng mL-1 of cathepsin S ad only one lower point for approx 0.05-0.1 ng mL-1. Even zero point is not presented. In this situation the authors’ conclusions about LOD and LOQ need in detailed justification. Please give average zero (blank) signal and its SD in the manuscript. Basing on these data, the LOD and LOQ values will be not only reported, but also substantiated. The actually given clarification is strange: «Standard deviation of blank is equal to 0.01 ng mL-1 for the Ag/Au biosensor and 0.022 ng mL-1 for the Au biosensor. LOD was calculated as 3SD and LOQ as 10 SD.» SD is measured in A.U. of SPRi signals, whereas LOD and LOQ are measured in ng mL-1. Parameters with different dimensions cannot be equal.
Points corresponding to a concentration of the analyte equal to 0 have been added, as recommended. At these points the analytical signal is equal to 1030±20 A.U. (i.e. SD) for the Ag/Au curves and 572±37 A.U. for the Au curves. Such a small error bar is hardly visible above a point circle. In our opinion, a potential reader will rather be interested in information on how low an analyte concentration it is possible to determine.
Question 19 was the following: «How the found values of cathepsin S accord to literature data about healthy persons and patients with ovarian cancer?». The authors have answered that «We have no data for such a comparison». However, even in the initial manuscript they said that «Cath S … This enzyme is an emerging marker for ovarian cancer». Thus, somebody compared earlier the Cath S levels for healthy persons and for patients with ovarian cancer and made the conclusion that cathepsin S is an oncomarker (that was then included in the manuscript). The reviewer is asking only to present data of this somebodies' study of Cath S levels.
We have added more citations concerning Cath S and various cancers [20-22 ] and have sightly modified the related text (lines 108-110)